# Grey-box Prompt Optimization and Fine-Tuning for Cloud-Edge LLM Agents

## Abstract

Large Language Models (LLMs) are transforming the landscape of generative AI, delivering groundbreaking performance across diverse tasks. Yet, their immense model sizes tether most LLMs to the cloud, posing challenges for tasks that demand processing private and proprietary data. In this paper, we introduce a grey-box prompt optimization and fine-tuning framework for cloud-edge LLMs-paving the way for a seamless, hybrid approach that merges the best of both private and public cloud environments. This framework not only boosts flexibility and scalability but also empowers users with heightened security and compliance, optimizing cost and performance. Beyond that, it ensures robust disaster recovery and business continuity through redundancy and smart workload distribution. At the heart of our solution is an efficient algorithm with guaranteed convergence, specifically tailored to the structure of the grey-box optimization problem. We rigorously analyze and derive its non-asymptotic convergence rate. Our extensive experiments reveal that sandwiched tuning-our novel fine-tuning method-delivers up to a 47.9% performance improvement over traditional methods across multiple tasks.

## 1 INTRODUCTION

Large Language Models (LLMs) have demonstrated unprecedented performance across a wide range of tasks, from natural language understanding (Karanikolas et al., 2023) to generative applications (Li et al., 2023), owing to their large-scale architectures and training on massive datasets (Raiaan et al., 2024). Traditional LLMs are typically hosted in the cloud due to their significant computational requirements (Wang et al., 2023), including extensive memory, storage, and processing power, making them impractical to run on local devices. However, cloud-hosted LLMs come with certain drawbacks, such as potential privacy and security risks when sensitive or proprietary data is transmitted to and processed by remote servers (McEnroe et al., 2022). Additionally, reliance on cloud infrastructure can lead to latency issues, especially for real-time applications where fast response times are critical. Furthermore, the continuous use of cloud-hosted LLMs can incur significant operational costs, particularly for applications that require constant access to large-scale models.

To address these challenges, we propose a cloud-edge LLM agent framework in this paper, which leverages the best of both cloud and edge computing. By offloading the resource-intensive model training (Zhang et al., 2024b) and large-scale processing tasks to the cloud, the framework ensures that scalability and computational efficiency are maintained, allowing users to benefit from the powerful capabilities of large language models. Meanwhile, edge devices handle tasks that require sensitive data processing or real-time interactions, ensuring that privacy is preserved by keeping sensitive data local and reducing the latency typically associated with cloud-only deployments (Zhang et al., 2024c). This hybrid approach not only enhances privacy and security by minimizing data transmission to the cloud but also allows for more personalized and context-aware applications, as edge devices can tailor LLM responses to specific user needs or local conditions. Furthermore, the framework optimizes resource utilization by distributing tasks intelligently between the cloud and edge, ensuring that each task is executed in the most appropriate environment, leading to improved performance, reduced bandwidth usage, and enhanced user experience in real-time applications.

However, while the cloud-edge framework offers flexibility, scalability, and enhanced security and privacy, integrating cloud and edge LLMs seamlessly to perform generation tasks remains a sig-

nificant challenge. One of the primary difficulties lies in coordinating the cloud-hosted and edge-deployed models, ensuring smooth collaboration without introducing delays or inconsistencies in task execution. In addition, existing cloud-edge collaboration frameworks for LLMs (Zhang et al., 2024a; Yang et al., 2024b; Hao et al., 2024; Yao et al., 2024; Ding et al., 2024) only involve collaboration during the inference stage. Due to the black-box nature of cloud-hosted LLMs (Li et al., 2024), which often restricts insight into their internal workings, it is particularly challenging to optimize prompts (Sabbatella et al., 2024) or perform fine-tuning (Lin et al., 2024) effectively, especially when the goal is to handle these operations entirely at the edge to safeguard data privacy. This limitation makes it difficult to tailor or adapt the LLM's performance to specific tasks without risking the exposure of sensitive data, posing a technical barrier to achieving fully private and efficient cloud-edge LLM interactions (Yan et al., 2024).

To this end, we propose a grey-box joint prompt optimization and fine-tuning framework for cloud-edge LLM agents, aimed at integrating the powerful capabilities of cloud-hosted LLMs with edge-deployed LLM agents for personalized applications. By leveraging a grey-box approach, we can partially access the cloud-hosted LLM's behavior through input-output evaluations, enabling us to perform prompt optimization and fine-tuning at the edge without compromising privacy. This method allows for personalized and adaptive use of LLMs in sensitive environments while maintaining the computational efficiency of cloud resources, offering a balance between performance and privacy in cloud-edge LLM architectures. Our main contributions can be summarized as follows.

- Apart from traditional cloud-hosted LLM architectures, this paper presents a cutting-edge cloud-edge LLM agent framework by harnessing the power of cloud scalability and cost-efficiency alongside the security and low-latency benefits of edge computing. This hybrid approach offers a balanced solution, optimizing performance, data privacy, and resource management. Additionally, we develop an innovative method that combines prompt optimization and fine-tuning, all through the lens of grey-box optimization. To the best of our knowledge, this is the *first grey-box* optimization-based approach for fine-tuning an LLM agent in a hybrid cloud environment.

- Building on the unique structure of the grey-box optimization problem, we develop a Sandwiched Tuning framework for cloud-edge LLM agents, featuring a memory-efficient Zeroth-Order Cutting Plane (ZoCP) algorithm designed specifically for edge deployment. This approach unlocks privacy-preserving, personalized fine-tuning directly on edge devices, bridging the gap between performance and data security. Furthermore, the decomposable nature of cutting planes could facilitate a distributed implementation of the framework, which may improve scalability and computational efficiency for large-scale cloud-edge deployments. We rigorously derive a non-asymptotic convergence rate that is independent of the number of optimization parameters for the ZoCP algorithm, highlighting its scalability on large-scale models.

- We have conducted extensive experiments on a variety of challenging tasks, including LLM task decomposition, tool use, and multi-turn dialogue, alongside natural language understanding tasks like text classification, multiple choice, and single-turn question answering, using LLMs with parameter sizes ranging from 0.5B to 8B as base models for edge agents. The results demonstrate that the proposed method significantly outperforms state-of-the-art approaches, with performance improvements as high as more than 40% in certain cases.

## 2 RELATED WORK

**Cloud-Edge Collaboration for LLMs.** Existing methods mainly emphasize cloud-edge collaboration specifically during the LLM inference phase, overlooking other stages that could benefit from more integrated approaches (Friha et al., 2024). These methods can be broadly divided into two categories, i.e., task assignment based methods (Zhang et al., 2024a; Yang et al., 2024b) and task correction based methods (Hao et al., 2024; Yao et al., 2024). By leveraging a collaborative framework, edge LLMs can take over inference tasks when the cloud-based LLM service is unavailable, ensuring continuous service for the user Ding et al. (2024). However, these methods are limited to using pre-trained language models for inference, with minimal research exploring cloud-edge collaboration for joint prompt optimization and fine-tuning. Another concept related to LLM cloud-

edge collaboration is Split Federated Learning (SFL), which outsources LLM components to remote servers. However, existing SFL approaches generally assume known structures and parameters for both cloud and edge models. Additionally, SFL typically requires transmission of activations and gradients, whereas this paper uses prompt-level text for more efficient communication. Lastly, SFL necessitates joint cloud-edge operation for both training and inference, hindering low-latency edge standalone inference.

**Prompt Optimization and Model Fine-Tuning.** Existing works on prompt optimization can be mainly categorized into parametric model-based approaches (Diao et al., 2022; Shum et al., 2023) and LLM-based methods (Pryzant et al., 2023; Yang et al., 2023; Cheng et al., 2023). LLM-based prompt optimization methods (Pryzant et al., 2023; Yang et al., 2023; Cheng et al., 2023) leverage the LLMs to generate prompts that are both effective and easily understandable by humans. Fine-tuning LLMs (Ding et al., 2023) can be primarily categorized into three groups, including partial approaches, re-parameterized approaches, and additive approaches (Xu et al., 2023). Partial approaches (Zaken et al., 2021) and re-parameterized methods (Hu et al., 2021) require creating a task-specific copy of the entire model for each downstream task (Lester et al., 2021). In contrast, additive approaches (Houlsby et al., 2019) achieve greater parameter sharing by introducing additional learnable parameters tailored to specific tasks, while keeping the original network's parameters fixed. Recently, the potential for jointly performing prompt optimization and fine-tuning has been explored, but current work remains limited to white-box scenarios (Soylu et al., 2024).

**Grey-box Optimization.** Unlike white-box or black-box optimization (Bajaj et al., 2021), grey-box optimization problems (Astudillo & Frazier, 2021) refer to optimization problems where the nested function involves both white-box and black-box functions. In particular, in nested optimization problems (Gergel et al., 2016), grey-box optimization occurs when the gradients of some optimization variables remain unknown. The zeroth-order optimization (ZOO) (Chen et al., 2017) offers a promising approach to handle gradient-free optimization, using function evaluations rather than gradients to tackle optimization problems. Liu et al. (2020b); Xu et al. (2020); Wang et al. (2020); Huang et al. (2022) focus Min-Max zeroth-order optimization problems with strongly-concave inner problems. Chen et al. (2023a) proposes a gradient-free method for nested optimization with a convex inner problem. However, these methods are not directly applicable to the problem at hand, as they do not account for the unique structural challenges involved.

# 3 JOINT PROMPT AND FINE TUNING VIA GREY-BOX OPTIMIZATION

The joint prompt and fine-tuning via grey-box optimization approach is termed as *Sandwiched Tuning* framework. As shown in Fig.(1), it comprises an edge LLM agent, a high-performance cloud-hosted LLM, and a lightweight adapter model. The cloud LLM and edge LLM agent can collaborate on specific tasks through distinct operational paradigms. For instance, the edge LL agent can function as a prompt optimizer, refining the prompt before sending it to the cloud LLM. Another edge component of the framework, the adapter model, is responsible for processing the cloud LLM's response by mapping it to a loss function value, thus establishing an end-to-end training loop for the entire framework. Through this framework, the cloud-hosted LLM and edge LLM agent can collaborate seamlessly to perform tasks, leveraging the adapter model to enable the supervision training and automated optimization of edge models' parameters.

After collaborative training with cloud-based LLMs, the edge LLM has the potential to approach near-parity with the capabilities of cloud LLMs. Users can then make trade-offs between performance, cost, and privacy. During the inference phase, the performance of Edge Standalone Inference Mode may be slightly lower than that of cloud-edge collaboration; however, it does not require access to cloud services, resulting in lower latency, reduced overhead, and enhanced protection of local data privacy.

## 3.1 PROBLEM FORMULATION

Let model functions $f(\cdot), g(\cdot), v(\cdot)$ denote respectively the input output relationships of edge LLM, the cloud-hosted LLM, and the adapter model. Let $\mathbf{x} \in \mathbb{R}^n$ and $\mathbf{y} \in \mathbb{R}^m$ denote the learnable parameters associated with $f(\cdot)$ and $v(\cdot)$, where $n$ and $m$ respectively represent the number of parameters. To reduce computational costs, it is common practice to train only a subset of key

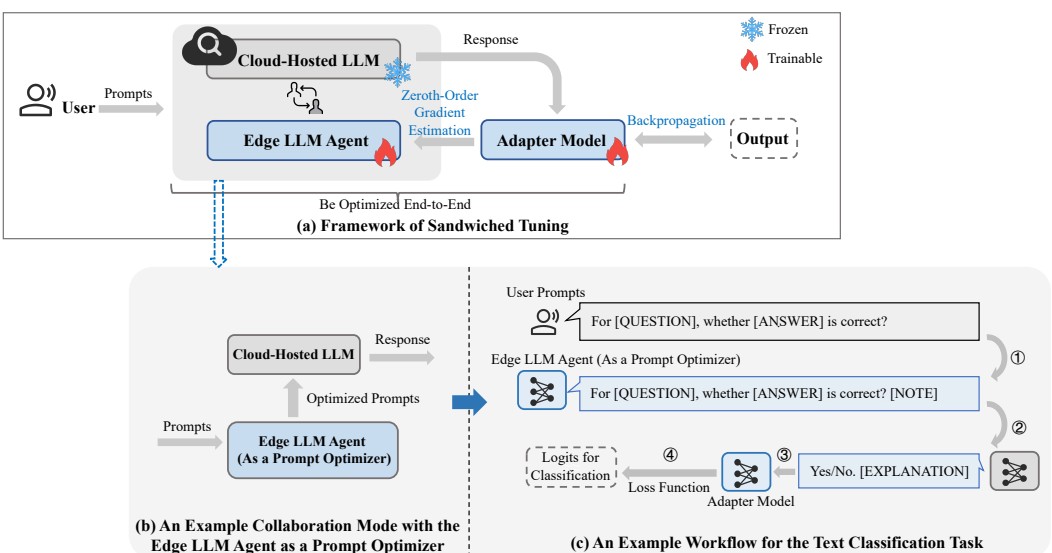

Figure 1: The Sandwiched Tuning framework.

parameters while keeping the remaining ones frozen. This approach minimizes resource usage while maintaining task-specific performance. Let $F(\mathbf{x}, \mathbf{y}) = L(v(g(f(\mathbf{x})), \mathbf{y}))$ denote the loss function of the downstream task, where $g(\cdot)$ is a black-box function. The overall process of the proposed sandwiched tuning is formulated as the following nested grey-box optimization problem:

$$\min_{\mathbf{x}} F(\mathbf{x}, \mathbf{y})$$
$$\text{s.t. } \mathbf{y} = \arg\min_{\mathbf{y}'} F(\mathbf{x}, \mathbf{y}'),$$
(1)

where the outer problem $\min_{\mathbf{x}} F(\mathbf{x}, \mathbf{y})$ aims to optimize the model parameters $\mathbf{x}$ of the edge LLM, and the inner optimization problem $\mathbf{y} = \arg\min_{\mathbf{y}'} F(\mathbf{x}, \mathbf{y}')$ aims to optimize the parameters $\mathbf{y}$ of the adapter model.

## 3.2 SANDWICHED TUNING

It is known that solving a nested optimization problem remains NP-hard (Kleinert et al., 2021). We employ a strategy that reformulates the nested zeroth-order optimization problem into a more tractable single-layer problem by incorporating the inner problem as a constraint within the outer optimization problem. To address the resulting formulation, we introduce a cutting plane method, inspired by Yang et al. (2014), which leverages zeroth-order gradient estimation techniques to efficiently solve the problem.

Specifically, denote $\varphi(\mathbf{x}) = \arg\min_{\mathbf{y}'} F(\mathbf{x}, \mathbf{y}')$ and $h(\mathbf{x}, \mathbf{y}) = ||\mathbf{y} - \varphi(\mathbf{x})||^2$. Consequently, the inner problem in Eq.(1) is equivalent to $h(\mathbf{x}, \mathbf{y}) = 0$, and we can regard the inner problem in Eq.(1) as a constraint of the outer problem to convert the original nested problem in Eq.(1) into the following single-level optimization problem:

$$\min_{\mathbf{x}, \mathbf{y}} F(\mathbf{x}, \mathbf{y})$$
$$\text{s.t. } h(\mathbf{x}, \mathbf{y}) = 0.$$
(2)

In Eq.(2), one important issue is how to calculate $\varphi(\mathbf{x})$ in function $h(\cdot)$. Since we are optimizing the parameters of neural networks, the function $F(\cdot)$ can be highly non-convex and it is therefore difficult to calculate $\arg\min_{\mathbf{y}'} F(\mathbf{x}, \mathbf{y}')$ exactly. As has been proven effective by previous work (Yang et al., 2021; Jiao et al., 2022), given the outer variable $\mathbf{x}$, we may approximate $\varphi(\mathbf{x})$ through stochastic gradient descent as:

$$\varphi(\mathbf{x}) = \mathbf{y}' - \eta_y \nabla_y \tilde{F}(\mathbf{x}, \mathbf{y}'; \mathcal{B}),$$
(3)

where $\mathcal{B}$ denotes a specific mini-batch of data samples drawn from the distribution $\mathcal{D}$, $\eta_y$ is the step size. $\tilde{F}(\cdot)$ represents the first-order Taylor approximation of $F(\cdot)$, that is, for a given point $\bar{\mathbf{x}}$, we have $\tilde{F}(\mathbf{x}, \mathbf{y}') = F(\bar{\mathbf{x}}, \mathbf{y}') + \nabla_{\mathbf{x}} F(\bar{\mathbf{x}}, \mathbf{y}')^\top (\mathbf{x} - \bar{\mathbf{x}})$. The existence of the black-box function $g(\cdot)$ prevents us from directly computing $\nabla_{\mathbf{x}} F(\bar{\mathbf{x}}, \mathbf{y}')$. Therefore, we further approximate $\nabla_{\mathbf{x}} F(\bar{\mathbf{x}}, \mathbf{y}')$ by stochastic samples and a zeroth-order gradient estimator (Liu et al., 2020a) as

$$\hat{\nabla}_{\mathbf{x}} F(\bar{\mathbf{x}}, \mathbf{y}'; \mathcal{B}) = \frac{F(\bar{\mathbf{x}} + \epsilon \mathbf{z}, \mathbf{y}'; \mathcal{B}) - F(\bar{\mathbf{x}} - \epsilon \mathbf{z}, \mathbf{y}'; \mathcal{B})}{2\epsilon} \mathbf{z}, \tag{4}$$

where $\mathbf{z}$ is randomly sampled with $\mathbf{z} \sim \mathcal{N}(0, \mathbf{I}_n)$, $\epsilon$ is the perturbation scale. The estimation in Eq.(4) can be averaged over $d$ sampled $\mathbf{z}$. We take $d = 1$ for the sake of efficiency.

According to Eq.(3) and Eq.(4), we can obtain a relaxed problem of Eq.(2) as:

$$\min_{\mathbf{x}, \mathbf{y}} F(\mathbf{x}, \mathbf{y})$$
$$\text{s.t. } h(\mathbf{x}, \mathbf{y}) \le \varepsilon, \tag{5}$$

where $\varepsilon > 0$ is a constant. Note that $h(\mathbf{x}, \mathbf{y})$ is convex w.r.t. $(\mathbf{x}, \mathbf{y})$ according to the fact that these operations preserve convexity (Boyd et al., 2004). Therefore, the feasible set of $h(\mathbf{x}, \mathbf{y}) \le \varepsilon$ is a convex set. To approximate this feasible set, we adopt a cutting plane method (Yang et al., 2014), which has been proven to be computationally efficient. The primary idea underlying the cutting-plane method is to approximate the optimal solution by introducing linear constraints (called cutting planes) iteratively into the feasible solution space of the target problem. These constraints form a linear relaxation of the problem, ensuring that the solutions remain within this relaxed polyhedron. Precisely, the polyhedron can be denoted as:

$$\mathcal{P} = \{\boldsymbol{a}_l^\top \mathbf{x} + \boldsymbol{b}_l^\top \mathbf{y} + c_l \le 0, \forall l \in [|\mathcal{P}|]\}, \tag{6}$$

where $\boldsymbol{a}_l \in \mathbb{R}^n$, $\boldsymbol{b}_l \in \mathbb{R}^m$, and $c_l \in \mathbb{R}^1$ are parameters of the $l^{th}$ cutting planes. $|\mathcal{P}| < p$ is the number of cutting planes. Then the problem in Eq.(5) can be approximated as follows:

$$\min_{\mathbf{x}, \mathbf{y}} F(\mathbf{x}, \mathbf{y})$$
$$\text{s.t. } \boldsymbol{a}_l^\top \mathbf{x} + \boldsymbol{b}_l^\top \mathbf{y} + c_l \le 0, \forall l \in [|\mathcal{P}|]. \tag{7}$$

The Lagrangian function of Eq.(7) is:

$$L_p(\mathbf{x}, \mathbf{y}, \{\lambda_l\}) = F(\mathbf{x}, \mathbf{y}) + \sum_{l=1}^{|\mathcal{P}|} \lambda_l (\boldsymbol{a}_l^\top \mathbf{x} + \boldsymbol{b}_l^\top \mathbf{y} + c_l), \tag{8}$$

where $\lambda_l \in \mathbb{R}^1$ is the dual variable. Then in the $(t+1)^{th}$ iteration, we update the parameters as follows:

$$\mathbf{x}^{t+1} = \mathbf{x}^t - \eta \hat{\nabla}_{\mathbf{x}} L_p(\mathbf{x}^t, \mathbf{y}^t, \{\lambda_l^t\}; \mathcal{B}), \tag{9}$$
$$\mathbf{y}^{t+1} = \mathbf{y}^t - \eta \nabla_{\mathbf{y}} L_p(\mathbf{x}^{t+1}, \mathbf{y}^t, \{\lambda_l^t\}; \mathcal{B}), \tag{10}$$
$$\lambda_l^{t+1} = \lambda_l^t + \eta \nabla_{\lambda_l} L_p(\mathbf{x}^{t+1}, \mathbf{y}^{t+1}, \{\lambda_l^t\}; \mathcal{B}), l = 1, \cdots, |\mathcal{P}^t|, \tag{11}$$

where $\mathcal{P}^t$ denotes the polyhedron in the $(t+1)^{th}$ iteration. $\eta$ is the step size. $\hat{\nabla}_{\mathbf{x}} L_p(\mathbf{x}^t, \mathbf{y}^t, \{\lambda_l^t\}; \mathcal{B})$ denotes the gradient estimated as:

$$\hat{\nabla}_{\mathbf{x}} L_p(\mathbf{x}^t, \mathbf{y}^t, \{\lambda_l^t\}; \mathcal{B}) = \frac{L_p(\mathbf{x}^t + \epsilon \mathbf{z}, \mathbf{y}^t, \{\lambda_l^t\}; \mathcal{B}) - L_p(\mathbf{x}^t - \epsilon \mathbf{z}, \mathbf{y}^t, \{\lambda_l^t\}; \mathcal{B})}{2\epsilon} \mathbf{z}, \tag{12}$$

where $\mathbf{z} \in \mathbb{R}^n$ is randomly sampled with $\mathbf{z} \sim \mathcal{N}(0, \mathbf{I}_n)$.

The last issue we are concerned with is *how to update the cutting planes*. Most existing cutting plane methods assume that all the parameters and gradients of the model are available and therefore cannot be directly applied to our problem. Denote the feasible region of the problem in Eq.(5) as $\mathcal{Z}$. If $(\mathbf{x}^{t+1}, \mathbf{y}^{t+1})$ is not feasible for Eq.(5), that is, $h(\mathbf{x}^{t+1}, \mathbf{y}^{t+1}) > \varepsilon$, we aim to find a cutting plane to separate $(\mathbf{x}^{t+1}, \mathbf{y}^{t+1})$ from $\mathcal{Z}$. Generally, a valid cutting plane satisfies the following:

$$\begin{cases} \boldsymbol{a}_l^\top \mathbf{x} + \boldsymbol{b}_l^\top \mathbf{y} + c_l \le 0, \forall (\mathbf{x}, \mathbf{y}) \in \mathcal{Z} \\ \boldsymbol{a}_l^\top \mathbf{x} + \boldsymbol{b}_l^\top \mathbf{y} + c_l > 0, \text{otherwise} \end{cases}. \tag{13}$$

Since we utilize a first-order Taylor approximation of $F(\cdot)$ and one-step gradient descent to approximate $\varphi(x)$, $h(\mathbf{x}, \mathbf{y})$ is a convex function. So we have:

$$h(\mathbf{x}, \mathbf{y}) \geq h(\mathbf{x}^{t+1}, \mathbf{y}^{t+1}) + \begin{bmatrix} \dfrac{\partial h(\mathbf{x}^{t+1}, \mathbf{y}^{t+1})}{\partial \mathbf{x}} \\ \dfrac{\partial h(\mathbf{x}^{t+1}, \mathbf{y}^{t+1})}{\partial \mathbf{y}} \end{bmatrix}^{\top} \left( \begin{bmatrix} \mathbf{x} \\ \mathbf{y} \end{bmatrix} - \begin{bmatrix} \mathbf{x}^{t+1} \\ \mathbf{y}^{t+1} \end{bmatrix} \right). \tag{14}$$

According to Eq.(13) and Eq.(14), it is obvious that we can find a new cutting plane $cp_{new}, \boldsymbol{a}_{new}^{\top}\mathbf{x} + \boldsymbol{b}_{new}^{\top}\mathbf{y} + c_{new} \leq 0$, as follows:

$$h(\mathbf{x}^{t+1}, \mathbf{y}^{t+1}) + \begin{bmatrix} \dfrac{\partial h(\mathbf{x}^{t+1}, \mathbf{y}^{t+1})}{\partial \mathbf{x}} \\ \dfrac{\partial h(\mathbf{x}^{t+1}, \mathbf{y}^{t+1})}{\partial \mathbf{y}} \end{bmatrix}^{\top} \left( \begin{bmatrix} \mathbf{x} \\ \mathbf{y} \end{bmatrix} - \begin{bmatrix} \mathbf{x}^{t+1} \\ \mathbf{y}^{t+1} \end{bmatrix} \right) \leq \varepsilon. \tag{15}$$

Precisely, the parameters of cutting planes are given by

$$\boldsymbol{a}_{new} = \frac{\partial h(\mathbf{x}^{t+1}, \mathbf{y}^{t+1})}{\partial \mathbf{x}}, \qquad \boldsymbol{b}_{new} = \frac{\partial h(\mathbf{x}^{t+1}, \mathbf{y}^{t+1})}{\partial \mathbf{y}},$$

$$c_{new} = h(\mathbf{x}^{t+1}, \mathbf{y}^{t+1}) - \begin{bmatrix} \dfrac{\partial h(\mathbf{x}^{t+1}, \mathbf{y}^{t+1})}{\partial \mathbf{x}} \\ \dfrac{\partial h(\mathbf{x}^{t+1}, \mathbf{y}^{t+1})}{\partial \mathbf{y}} \end{bmatrix}^{\top} \begin{bmatrix} \mathbf{x}^{t+1} \\ \mathbf{y}^{t+1} \end{bmatrix} - \varepsilon, \tag{16}$$

where $\frac{\partial h(\mathbf{x}^{t+1}, \mathbf{y}^{t+1})}{\partial \mathbf{y}}$ and $\frac{\partial h(\mathbf{x}^{t+1}, \mathbf{y}^{t+1})}{\partial \mathbf{x}}$ can be calculated according to Eq.(3) and Eq.(4). Then we add $cp_{new}$ to the polytope $\mathcal{P}^{t+1}$ and add $\lambda_{new}$ to the set $\{\lambda_l^{t+1}\}$. Note that we update the cutting planes every $k$ iteration and the inactive cutting planes, whose dual variable is less than a threshold for some successive iterations, will be deleted to save computing resources. The details of the proposed ZoCP algorithm are summarized in Algorithm 1.

---

**Algorithm 1** Zeroth-order Cutting Plane (ZoCP) Algorithm

---

**Initialization:** iteration $t = 0$, trainable parameters of the edge LLM $\mathbf{x}^0$, trainable parameters of the adapter model $\mathbf{y}^0$, dual parameters $\{\lambda_l^0\}$, polytope $\mathcal{P}^0$.
**while** not terminated **do**
    Update parameters $\mathbf{x}^{t+1}$, $\mathbf{y}^{t+1}$, and $\{\lambda_l^{t+1}\}$ according to Eq.(9), Eq.(10), and Eq.(11).
    **if** $(t + 1)$ mod $k == 0$ **then**
      **if** $h(\mathbf{x}^{t+1}, \mathbf{y}^{t+1}) > \varepsilon$ **then**
        Find new cutting plane according to Eq.(16) and update $\mathcal{P}^{t+1}$ and $\{\lambda_l^{t+1}\}$.
      **end if**
    **end if**
**end while**

---

### 3.3 NON-ASYMPTOTIC CONVERGENCE ANALYSIS

In this section, we demonstrate that ZoCP is guaranteed to converge and carry out non-asymptotic analysis in Theorem 1, focusing on quantifying how quickly this algorithm approaches an optimal solution within a specified number of iterations or time steps, providing concrete guarantees for performance over a finite number of steps. This type of analysis is particularly useful in practical applications where resource constraints, such as time or computational power, limit the number of iterations an algorithm can run. Following previous works (Malladi et al., 2023; Ling et al., 2024; Chen et al.), under the Assumptions on smoothness and Local r-effective rank of the $L_p(\mathbf{x}^t, \mathbf{y}^t, \{\lambda_l^t\})$ function, we establish the following convergence rate for ZoCP.

**Theorem 1.** *(Non-asymptotic Convergence Analysis) The optimal value of the objective function in the approximate problem Eq. (7) converges monotonically when the number of cutting planes*

*increases progressively. In addition, under Assumptions on smoothness and Local r-effective rank of the $L_p(\mathbf{x}^t, \mathbf{y}^t, \{\lambda_l^t\})$ function (Malladi et al., 2023), ZoCP achieves $\mathbb{E}[L_p(\mathbf{x}^t, \mathbf{y}^t, \{\lambda_l^t\})] \leq L_p^* + \epsilon$ after*

$$t = \mathcal{O}((\frac{r}{d} + 1)(\frac{1}{p})(\frac{L}{\mu} + \frac{L\alpha}{\mu^2 B}) \log \frac{L_p(\mathbf{x}^0, \mathbf{y}^0, \{\lambda_l^0\}) - L_p^*}{\epsilon}) \tag{17}$$

*iterations. The proof of Theorem 1 is given in Appendix A.1.*

## 4 EXPERIMENTS

In this section, we first validate the effectiveness of the proposed framework in the following complex real-world scenarios: 1) LLM task decomposition, 2) tool use, and 3) multi-turn dialogue. We also examine the performance of the proposed sandwiched tuning framework over a diverse set of natural language understanding (NLU) tasks, including text classification, multiple-choice, and single-turn question answering. Our experiments use Qwen-max (Bai et al., 2023) as the cloud-hosted LLM. The base LLMs of the edge agent considered in our experiments include Qwen2.5-0.5B (Team, 2024), GPT-2 (1.5B) (Radford et al., 2019), Qwen2-7B (Yang et al., 2024a) and Llama3-8B (Touvron et al., 2023). The edge LLMs are deployed on 1 NVIDIA A100 GPU and 2 NVIDIA GeForce RTX 4090 GPUs for different experiments. It is worth mentioning that using a GPU on the edge side is not necessary.

### 4.1 LLM TASK DECOMPOSITION

Task decomposition is the process of breaking down complex tasks into more specific subtasks or task steps. The decomposition process requires LLMs to employ sophisticated semantic reasoning and precise text generation to effectively break tasks down into manageable subtasks that can be more easily solved.

In our experiments, we utilize GPT-2, Qwen2-7B, and Llama3-8B as the base models of edge agents, while a BERT-Mini as the adapter model. The cloud-hosted LLM and the edge agent collaborate by independently performing task decomposition and sharing insights, enhancing their overall problem-solving capabilities. The experiments are conducted on the Orca-Math-200K (Mitra et al., 2024) and TaskLAMA (Yuan et al., 2024) datasets to evaluate the effectiveness of the proposed framework.

Table 1: Performance Comparison on LLM Task Decomposition.

| Model | GPT2 | | GPT2 (Optimized) | | Qwen2-7B | | Qwen2-7B (Optimized) | | Llama3-8B | | Llama3-8B (Optimized) | |
|---|---|---|---|---|---|---|---|---|---|---|---|---|
| | F1 | SIM | F1 | SIM | F1 | SIM | F1 | SIM | F1 | SIM | F1 | SIM |
| Orca-Math-200K | 18.4 | 72.4 | 18.6 | 72.9 | 37.9 | 85.7 | 39.4 | 88.0 | _50.5_ | _91.7_ | **51.9** | **92.2** |
| TaskLAMA | 3.7 | 65.3 | 3.8 | 65.4 | 25.3 | 87.6 | 27.9 | _89.4_ | _32.6_ | 88.3 | **34.7** | **90.1** |

The performance comparison results are presented in Table 1. The best results are highlighted in bold, and the second-best method is underlined. F1 score (F1) and cosine similarity (SIM) are used to reflect the gap in task decomposition capabilities between the cloud-hosted LLM and the edge agent. Experiments have shown that cloud-hosted LLMs can improve edge agents' performance in task decomposition, with model size significantly influencing outcomes. Both Qwen2-7B and Llama3-8B exhibited notable improvements after optimization. The results suggest that for complex tasks like LLM task decomposition, effective performance is only achievable when model parameters reach a certain scale, as demonstrated by GPT-2's clear performance gap compared to larger models. This aligns with LLM scaling laws, indicating a size threshold necessary for handling specific tasks.

### 4.2 TOOL USE

Although LLMs perform well on complex natural language tasks, they may struggle with simpler tasks that humans handle easily, such as character counting, resulting in high error rates. Recent advances in tool utilization have shown promise in enhancing LLM capabilities (Qu et al., 2024).

To validate the proposed approach, we evaluated it on three representative tasks: floating-point arithmetic, mathematical comparison, and character counting.

We used GPT-2 (1.5B) (Radford et al., 2019) as the edge agent, and Qwen2.5-0.5B (Team, 2024) as the adapter model. We conducted experiments under three different settings. a) `Cloud-Only`: The cloud LLM independently infers the answer. b) `Sandwiched Tuning`. In an edge-cloud collaborative framework, the cloud LLM is given an optimized prompt to rephrase the question or generate a solution formula. The edge agent then uses specific tools (e.g., calculator, floating-point comparison, character counting) to complete the task. c) `Sandwiched Tuning-Edge`. After fine-tuning, only the edge agent is used, leveraging optimized prompts and tool capabilities for task execution. We computed the success rate (Zhuang et al., 2023) between ground-truth answers and predicted answers.

For datasets, we used a publicly available dataset (APE-210k) for the mathematical reasoning task (Zhao et al., 2020). We further created three datasets: "Float-Arithmetic" for real-world floating-point problems, "Float-Comparison" for comparing two floating numbers, and "character counting" for counting specific characters in a string. More details are provided in the Appendix A.2.3.

Table 2: Success Rate of Tool Use on Different Settings.

|  | Cloud-Only | Sandwiched Tuning | Sandwiched Tuning-Edge |
|---|---|---|---|
| Float-Arithmetic | 0.580 | **0.795** | 0.095 |
| APE-210k | 0.525 | **0.725** | 0.145 |
| Float-Comparison | 0.830 | **0.967** | 0.950 |
| Character-Counting | 0.670 | **0.991** | 0.972 |

The results in Table 2 show that the sandwiched tuning framework significantly improves performance in challenging tasks where LLMs typically perform poorly. By leveraging an edge-cloud collaborative setup, our method enhances performance in both complex mathematical calculations and simpler tasks. Specifically, the sandwiched tuning framework achieved the highest success rate improvement of up to *47.9%* in individual tasks, with an average improvement of *33.8%* compared against the cloud-only setup. It is noteworthy that, in the float-arithmetic task, which requires high reasoning capabilities, the sandwiched tuning framework boosted success rate from 9.5% (edge-only) to 79.5% by combining the cloud model's reasoning abilities with the edge model's tool utilization capabilities.

Table 3: Latency (s) of Float Arithmetic Task.

|  | Cloud-Only | Sandwiched Tuning | Sandwiched Tuning-Edge |
|---|---|---|---|
| Float-Arithmetic | 27.1 | **19.5** | 4.5 |

Furthermore, as shown in Table 3, the proposed cloud-edge architecture can significantly reduce the latency in the floating-arithmetic task. Although Sandwiched Tuning increases the communication overhead for transmitting data from the cloud LLM to the edge LLM compared to Cloud-Only methods, the edge LLM accelerates task inference by invoking tools, thereby significantly reducing the overall latency. More experiment results on tradeoffs among cloud-edge load distribution, inference latency, and inference accuracy can be found in Appendix A.2.3.

### 4.3 MULTI-TURN DIALOGUE GENERATION

Generating high-quality dialogues presents significant challenges, especially in ensuring contextual relevance and coherence in conversations. Traditional LLMs such as GPT-4 and GPT-3.5 perform well in general, but their dialogue generation can be improved by incorporating relevant conversation history. Ensuring that the generated responses remain consistent with prior exchanges, while also providing new, accurate, and relevant information, is a complex task that requires careful selection of dialogue examples from historical data.

In this task, the goal is to generate high-quality dialogues via incorporating relevant conversation history. We conducted experiments across 6 customer support datasets (derived from Twitter interactions (Axelbrooke, 2017)) to assess the effectiveness of our proposed framework in generating more accurate and contextually relevant dialogues. We compare 2 strategies in selecting conversation examples, including `Random`, which selects dialogue samples without any specific optimization; and `ICL`, which retrieves 5 examples and randomly selects 2 for generation. For our method, `Sandwiched Tuning`, which also retrieves 5 dialogues but utilizes an edge agent to determine the 2 most relevant ones. We report the "Win Rate" used in Dubois et al. (2024) across six datasets, with the score reflecting how often each method generates higher-quality dialogues compared to its competitors.

Table 4: Results of Dialogue Generation Quality.

| Datasets | Methods | | |
|---|---|---|---|
| | Random | ICL | Sandwiched Tuning |
| Hulu_Support | 0.785 | 0.843 | **0.864** |
| Sainsburys | 0.680 | 0.765 | **0.782** |
| Comcastcares | 0.744 | 0.762 | **0.816** |
| Sprintcare | 0.686 | 0.713 | **0.761** |
| UPSHelp | 0.569 | 0.616 | **0.639** |
| XboxSupport | 0.699 | 0.732 | **0.754** |
| AVG | 0.694 | 0.739 | **0.769** |

The results, as shown in Table 4, demonstrate that the `Sandwiched Tuning` method consistently surpasses both `Random` and `ICL`. Notably, our method achieves up to *7.1%* improvement over `ICL`. By utilizing an edge agent to intelligently select the most contextually relevant dialogue samples, our method ensures that the generated conversations are not only accurate but also closely aligned with the user's question, without the need to construct specific dialogue states or predefined workflows.

## 4.4 NATURAL LANGUAGE UNDERSTANDING TASKS

In this section, we evaluate the effectiveness of our framework on a diverse set of NLU tasks. Table 5 presents the comparison results against baseline methods on text classification, multiple-choice questions answering (MCQA), and single-turn question answering tasks. The best results are highlighted in bold, and the second-best method is underlined. We use `Manual Prompting`, `Zero-shot CoT` (Kojima et al., 2022), `Random ICL`, and `OPRO` (Yang et al., 2023) as baseline methods. The above prompt optimization baseline methods all use Qwen-max as the cloud LLM. Detailed information regarding datasets, prompt templates, baselines, and experimental configurations can be found in Appendix A.2.1.

The proposed sandwiched tuning method consistently achieves superior performance across most datasets. Notably, the GPT-2 variant outperforms the second-best baseline by margins ranging from 1% to 43% (on the SQuAD dataset). These results validate the efficacy of the sandwiched tuning framework, which leverages grey-box optimization to jointly perform prompt optimization and fine-tuning. Besides, the LLM-based prompt optimization methods (`Sandwiched Tuning` and `OPRO`) generally perform better than heuristic methods thanks to the semantic understanding capabilities of LLMs. The heuristic prompt optimization methods are not stable, they may achieve good performance in some scenarios but perform badly at other times. It can also be observed that the performance of our framework improves with parameter size of the edge LLM agent, which aligns with the Scaling Laws of LLMs.

## 4.5 ABLATION STUDY

We conduct ablation experiments and the results are shown in Table 6. `ST-Prompt` denotes a stripped-down version of sandwiched tuning that only optimizes the edge LLM agent, and `ST-Adapter` only optimizes the adapter model. `Sandwiched Tuning` outperforms all its stripped-down versions in our experiments. Consistent with our motivation, the edge LLM agent

Table 5: Performance comparison on NLU tasks.

| Method | Manual Prompt | Zero-shot CoT | Random ICL | OPRO | Ours (GPT-2) | Ours (Qwen2-7B) | Ours (Llama3-8B) |
|---|---|---|---|---|---|---|---|
| Text Classification (Accuracy) | | | | | | | |
| SST-2 | 0.714 | 0.869 | 0.688 | 0.879 | 0.888 | 0.895 | **0.920** |
| MRPC | 0.733 | 0.800 | 0.787 | 0.853 | 0.832 | 0.876 | **0.884** |
| Tweets_Hate | 0.924 | 0.908 | 0.836 | 0.947 | 0.956 | 0.960 | **0.980** |
| Wiki_Toxic | 0.556 | 0.764 | 0.336 | 0.849 | 0.872 | 0.912 | **0.924** |
| FELM | 0.588 | 0.452 | 0.336 | 0.560 | 0.708 | 0.728 | **0.780** |
| BoolQ | 0.879 | 0.870 | 0.880 | 0.876 | 0.900 | 0.908 | **0.960** |
| WiC | 0.702 | 0.668 | 0.705 | 0.713 | 0.730 | 0.732 | **0.736** |
| MCQA (Accuracy) | | | | | | | |
| COPA | 0.936 | 0.844 | 0.941 | 0.948 | 0.960 | 0.984 | **0.988** |
| SWAG | 0.696 | 0.708 | 0.676 | 0.760 | 0.768 | 0.780 | **0.792** |
| Single-Turn Question Answering (F1 Score) | | | | | | | |
| SQuAD | 0.330 | 0.317 | 0.641 | 0.584 | 0.832 | 0.840 | **0.897** |
| DROP | 0.185 | 0.144 | 0.385 | 0.203 | 0.472 | 0.485 | **0.502** |

facilitates the model's understanding of human intentions, and the adapter model enhances adaptation to downstream tasks.

Table 6: Ablation Study of Different Components of the Sandwiched Tuning Framework.

| | ST-Prompt | ST-Adapter | Sandwiched Tuning (GPT-2) |
|---|---|---|---|
| SST-2 | 0.860 | 0.850 | **0.888** |
| MRPC | 0.800 | 0.784 | **0.832** |
| Tweets_Hate | 0.940 | 0.926 | **0.956** |
| Wiki_Toxic | 0.890 | 0.880 | **0.912** |
| FELM | 0.700 | 0.694 | **0.708** |
| BoolQ | 0.752 | 0.746 | **0.768** |
| WiC | 0.712 | 0.704 | **0.730** |

## 5 CONCLUSION

The grey-box prompt optimization and fine-tuning framework introduced in this paper provides a transformative solution for cloud-edge LLMs, addressing key challenges in balancing security, scalability, and performance. By leveraging a hybrid approach, the proposed framework allows for the secure processing of private data while taking advantage of the computational power of cloud-hosted LLMs. The proposed sandwiched tuning algorithm, with its guaranteed non-asymptotic convergence, ensures efficient optimization tailored to the joint prompt optimization and fine-tuning problem. The extensive experimental results demonstrate the superiority of our sandwiched tuning method, delivering substantial performance improvements of up to 47.9% over traditional methods. We hope this work paves the way for more flexible and resilient LLM deployment and tuning, offering a promising path forward for applications requiring both privacy-preserving and high-performance LLM deployment and tuning solutions.

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
