# A APPENDIX

## A.1 PROOF OF THEOREM 1

*Proof.* Since cutting planes are generated and added to the polytope every $k$ iteration, the polytope $\mathcal{P}$ satisfies that $\mathcal{P}^0 \supseteq \mathcal{P}^k \supseteq \cdots \mathcal{P}^{nk}$. Known that the feasible region of the problem in Eq.(5) is $\mathcal{Z}$, we denote the feasible region of Eq.(7) in $k^{th}$ iteration as $\mathcal{Z}'^k$. Then we have $\mathcal{Z}'^0 \supseteq \mathcal{Z}'^k \supseteq \cdots \mathcal{Z}'^{nk} \supseteq \mathcal{Z}$. Denoting the optimal value of the objective function in Eq.(7) at $k^{th}$ iteration as $F(\mathbf{x}^{k*}, \mathbf{y}^{k*})$, we can obtain that:

$$F(\mathbf{x}^{0*}, \mathbf{y}^{0*}) \le F(\mathbf{x}^{k*}, \mathbf{y}^{k*}) \le \cdots \le F(\mathbf{x}^{n*}, \mathbf{y}^{n*}). \tag{18}$$

Subsequently, we have that

$$\frac{F^*}{F(\mathbf{x}^{0*}, \mathbf{y}^{0*})} \ge \frac{F^*}{F(\mathbf{x}^{k*}, \mathbf{y}^{k*})} \ge \cdots \ge \frac{F^*}{F(\mathbf{x}^{nk*}, \mathbf{y}^{nk*})} \ge \alpha, \tag{19}$$

where $F^*$ denotes the optimal objective value of the problem in Eq.(5), $\alpha \ge 1$. It can be observed that $\frac{F^*}{F(\mathbf{x}^{k*}, \mathbf{y}^{k*})}$ is a monotonically nonincreasing sequence. Therefore, when $nk \to \infty$, the optimal objective value of the problem in Eq.(7) will converge to $\alpha$ monotonically.

According to Eq.(12), in the $\epsilon \to 0$ limit, we have

$$\hat{\nabla}_{\mathbf{x}} L_p(\mathbf{x}, \mathbf{y}, \{\lambda_l\}; \mathcal{B}) = \frac{1}{Bd} \sum_{\xi \in \mathcal{B}} \sum_{i \in [d]} \mathbf{z}_i \mathbf{z}_i^\top \nabla_{\mathbf{x}} L_p(\mathbf{x}, \mathbf{y}, \{\lambda_l\}; \xi), \tag{20}$$

and $\mathbb{E}[\hat{\nabla}_{\mathbf{x}} L_p(\mathbf{x}, \mathbf{y}, \{\lambda_l\}; \mathcal{B})] = \nabla_{\mathbf{x}} L_p(\mathbf{x}, \mathbf{y}, \{\lambda_l\})$. That is, $\hat{\nabla}_{\mathbf{x}} L_p(\mathbf{x}, \mathbf{y}, \{\lambda_l\}; \mathcal{B})$ is an unbiased estimator of the gradient.

The second moment can be computed as

$$\mathbb{E}[\hat{\nabla}_{\mathbf{x}} L_p(\mathbf{x}, \mathbf{y}, \{\lambda_l\}; \mathcal{B}) \hat{\nabla}_{\mathbf{x}} L_p(\mathbf{x}, \mathbf{y}, \{\lambda_l\}; \mathcal{B})^\top]$$
$$= \frac{1}{B^2 d^2} \sum_{\xi_1, \xi_2 \in \mathcal{B}} \sum_{i,j \in [d]} \mathbb{E}[(\mathbf{z}_i \mathbf{z}_i^\top \nabla_{\mathbf{x}} L_p(\mathbf{x}, \mathbf{y}, \{\lambda_l\}; \xi_1))(\mathbf{z}_j \mathbf{z}_j^\top \nabla_{\mathbf{x}} L_p(\mathbf{x}, \mathbf{y}, \{\lambda_l\}; \xi_2))^\top]. \tag{21}$$

Given two arbitrary vectors $\mathbf{u}$ and $\mathbf{v}$, we can obtain

$$\mathbb{E}_{\mathbf{z}_i, \mathbf{z}_j}[\mathbf{z}_i \mathbf{z}_i^\top \mathbf{u} \mathbf{v}^\top \mathbf{z}_j \mathbf{z}_j^\top] = \mathbf{u} \mathbf{v}^\top, i \ne j, \tag{22}$$

and

$$\mathbb{E}_{\mathbf{z}_i}[\mathbf{z}_i \mathbf{z}_i^\top \mathbf{u} \mathbf{v}^\top \mathbf{z}_i \mathbf{z}_i^\top] = \mathbb{E}_{\mathbf{z}}[\mathbf{z}^{\otimes 4}](\mathbf{u}, \mathbf{v}) = \frac{3n}{n+2} \mathrm{Sym}(\mathbf{I}^{\otimes 2})(\mathbf{u}, \mathbf{v}) = \frac{n}{n+2} \mathbf{u}^\top \mathbf{v} \mathbf{I} + \frac{2n}{n+2} \mathbf{u} \mathbf{v}^\top. \tag{23}$$

It follows that

$$\mathbb{E}[\hat{\nabla}_{\mathbf{x}} L_p(\mathbf{x}, \mathbf{y}, \{\lambda_l\}; \mathcal{B}) \hat{\nabla}_{\mathbf{x}} L_p(\mathbf{x}, \mathbf{y}, \{\lambda_l\}; \mathcal{B})^\top]$$
$$= \frac{1}{B^2} \sum_{\xi_1, \xi_2 \in \mathcal{B}} (\frac{d-1}{d} + \frac{2n}{d(n+2)}) \mathbb{E}[L_p(\mathbf{x}, \mathbf{y}, \{\lambda_l\}; \xi_1) L_p(\mathbf{x}, \mathbf{y}, \{\lambda_l\}; \xi_2)^\top]$$
$$+ \frac{n}{d(n+1)} \mathbb{E}[L_p(\mathbf{x}, \mathbf{y}, \{\lambda_l\}; \xi_1)^\top L_p(\mathbf{x}, \mathbf{y}, \{\lambda_l\}; \xi_2)] \mathbf{I} \tag{24}$$
$$= (1 + \frac{n-2}{d(n+2)}) (\nabla_{\mathbf{x}} L_p(\mathbf{x}, \mathbf{y}, \{\lambda_l\}) \nabla_{\mathbf{x}} L_p(\mathbf{x}, \mathbf{y}, \{\lambda_l\})^\top + \frac{1}{B} \mathbf{\Sigma}_{\mathbf{x}}(\mathbf{x}, \mathbf{y}, \{\lambda_l\}))$$
$$+ \frac{n}{d(n+2)} \mathbf{I}(||\nabla_{\mathbf{x}} L_p(\mathbf{x}, \mathbf{y}, \{\lambda_l\})||^2 + \frac{1}{B} \mathrm{tr}(\mathbf{\Sigma}_{\mathbf{x}}(\mathbf{x}, \mathbf{y}, \{\lambda_l\}))).$$

According to Eq.(24), we can obtain that

$$\mathbb{E}[||\hat{\nabla}_{\mathbf{x}} L_p(\mathbf{x}, \mathbf{y}, \{\lambda_l\}; \mathcal{B})||^2] = \frac{n+d-1}{d} \mathbb{E}[\nabla_{\mathbf{x}} L_p(\mathbf{x}, \mathbf{y}, \{\lambda_l\}; \mathcal{B})]. \tag{25}$$

By Taylor's theorem with remainder, we have

$$
\begin{aligned}
&L_p(\mathbf{x}^{t+1}, \mathbf{y}^t, \{\lambda_l^t\}) \\
=&L_p(\mathbf{x}^t, \mathbf{y}^t, \{\lambda_l^t\}) + \nabla_{\mathbf{x}} L_p(\mathbf{x}^t, \mathbf{y}^t, \{\lambda_l^t\})^\top (\mathbf{x}^{t+1} - \mathbf{x}^t) \\
&+ \int_0^1 \beta(\mathbf{x}^{t+1} - \mathbf{x}^t)^\top \nabla_{\mathbf{x}}^2 L_p(\beta \mathbf{x}^{t+1} + (1-\beta)\mathbf{x}^t, \mathbf{y}^t, \{\lambda_l^t\})(\mathbf{x}^{t+1} - \mathbf{x}^t)^\top d\beta.
\end{aligned}
\tag{26}
$$

According to the update rules of $\mathbf{x}$ and properties of $\{\mathbf{z}\}$, we have

$$
\begin{aligned}
||\mathbf{x}^{t+1} - \mathbf{x}^t|| &= \eta ||\hat{\nabla}_{\mathbf{x}} L_p(\mathbf{x}^t, \mathbf{y}^t, \{\lambda_l^t\}; \mathcal{B})|| \\
&\leq \frac{\eta\sqrt{n}}{Bd} \sum |\mathbf{z}_i^\top \nabla_{\mathbf{x}} L_p(\mathbf{x}^t, \mathbf{y}^t, \{\lambda_l^t\}; \xi)| \\
&\leq \eta n G(\mathbf{x}^t, \mathbf{y}^t, \{\lambda_l^t\}).
\end{aligned}
\tag{27}
$$

According to assumptions on smoothness and r-effective rank of the $L_p$ function and Eq.(27), we can obtain that

$$
\begin{aligned}
&L_p(\mathbf{x}^{t+1}, \mathbf{y}^t, \{\lambda_l^t\}) \\
\leq&L_p(\mathbf{x}^t, \mathbf{y}^t, \{\lambda_l^t\}) + \nabla_{\mathbf{x}} L_p(\mathbf{x}^t, \mathbf{y}^t, \{\lambda_l^t\})^\top (\mathbf{x}^{t+1} - \mathbf{x}^t) + (\mathbf{x}^{t+1} - \mathbf{x}^t)^\top \mathbf{H}_{\mathbf{x}}(\mathbf{x}^t, \mathbf{y}^t, \{\lambda_l^t\})(\mathbf{x}^{t+1} - \mathbf{x}^t) \\
=&L_p(\mathbf{x}^t, \mathbf{y}^t, \{\lambda_l^t\}) - \eta \nabla_{\mathbf{x}} L_p(\mathbf{x}^t, \mathbf{y}^t, \{\lambda_l^t\})^\top \hat{\nabla}_{\mathbf{x}} L_p(\mathbf{x}^t, \mathbf{y}^t, \{\lambda_l^t\}; \mathcal{B}) \\
&+ \frac{1}{2}\eta^2 \hat{\nabla}_{\mathbf{x}} L_p(\mathbf{x}^t, \mathbf{y}^t, \{\lambda_l^t\}; \mathcal{B})^\top \mathbf{H}_{\mathbf{x}}(\mathbf{x}^t, \mathbf{y}^t, \{\lambda_l^t\}) \hat{\nabla}_{\mathbf{x}} L_p(\mathbf{x}^t, \mathbf{y}^t, \{\lambda_l^t\}; \mathcal{B}).
\end{aligned}
\tag{28}
$$

Plugging Eq.(24) into Eq.(28) and taking the expectation to have

$$
\begin{aligned}
&\mathbb{E}[L_p(\mathbf{x}^{t+1}, \mathbf{y}^t, \{\lambda_l^t\})] \\
\leq&L_p(\mathbf{x}^t, \mathbf{y}^t, \{\lambda_l^t\}) - \eta ||\nabla_{\mathbf{x}} L_p(\mathbf{x}^t, \mathbf{y}^t, \{\lambda_l^t\})||^2 \\
&+ \frac{\eta^2}{2} \langle \mathbf{H}_{\mathbf{x}}(\mathbf{x}^t, \mathbf{y}^t, \{\lambda_l^t\}), \mathbb{E}[\hat{\nabla}_{\mathbf{x}} L_p(\mathbf{x}^t, \mathbf{y}^t, \{\lambda_l^t\}; \mathcal{B}) \hat{\nabla}_{\mathbf{x}} L_p(\mathbf{x}^t, \mathbf{y}^t, \{\lambda_l^t\}; \mathcal{B})^\top] \rangle \\
=&L_p(\mathbf{x}^t, \mathbf{y}^t, \{\lambda_l^t\}) - \eta ||\nabla_{\mathbf{x}} L_p(\mathbf{x}^t, \mathbf{y}^t, \{\lambda_l^t\})||^2 \\
&+ \frac{\eta^2}{2} \cdot \frac{n}{d(n+2)} (||\nabla_{\mathbf{x}} L_p(\mathbf{x}^t, \mathbf{y}^t, \{\lambda_l^t\})||^2 + \frac{1}{B}\mathrm{tr}(\boldsymbol{\Sigma}_{\mathbf{x}}(\mathbf{x}^t, \mathbf{y}^t, \{\lambda_l^t\}))) \mathrm{tr}(\mathbf{H}_{\mathbf{x}}(\mathbf{x}^t, \mathbf{y}^t, \{\lambda_l^t\})) \\
&+ \frac{\eta^2}{2} (1 + \frac{n-2}{d(n+2)})(\nabla_{\mathbf{x}} L_p(\mathbf{x}^t, \mathbf{y}^t, \{\lambda_l^t\})^\top \mathbf{H}_{\mathbf{x}}(\mathbf{x}^t, \mathbf{y}^t, \{\lambda_l^t\}) \nabla_{\mathbf{x}} L_p(\mathbf{x}^t, \mathbf{y}^t, \{\lambda_l^t\}) \\
&+ \frac{1}{B} \langle \boldsymbol{\Sigma}_{\mathbf{x}}(\mathbf{x}^t, \mathbf{y}^t, \{\lambda_l^t\}), \mathbf{H}_{\mathbf{x}}(\mathbf{x}^t, \mathbf{y}^t, \{\lambda_l^t\}) \rangle).
\end{aligned}
\tag{29}
$$

Assumptions on smoothness and r-effective rank of the $L_p$ function indicate that $||\mathbf{H}_{\mathbf{x}}(\mathbf{x}^t, \mathbf{y}^t, \{\lambda_l^t\})||_{op} \leq L$ and $\mathrm{tr}(\mathbf{H}_{\mathbf{x}}(\mathbf{x}^t, \mathbf{y}^t, \{\lambda_l^t\})) \leq Lr$. Thus, according to Eq.(29), we have

$$
\begin{aligned}
&\mathbb{E}[L_p(\mathbf{x}^{t+1}, \mathbf{y}^t, \{\lambda_l^t\})] \\
\leq&L_p(\mathbf{x}^t, \mathbf{y}^t, \{\lambda_l^t\}) - \eta ||\nabla_{\mathbf{x}} L_p(\mathbf{x}^t, \mathbf{y}^t, \{\lambda_l^t\})||^2 \\
&+ \frac{\eta^2 L}{2} (\frac{nr + n - 2}{d(n+2)} + 1)(||\nabla_{\mathbf{x}} L_p(\mathbf{x}^t, \mathbf{y}^t, \{\lambda_l^t\})||^2 + \frac{1}{B}\mathrm{tr}(\boldsymbol{\Sigma}_{\mathbf{x}}(\mathbf{x}^t, \mathbf{y}^t, \{\lambda_l^t\}))) \\
=&L_p(\mathbf{x}^t, \mathbf{y}^t, \{\lambda_l^t\}) - \eta ||\nabla_{\mathbf{x}} L_p(\mathbf{x}^t, \mathbf{y}^t, \{\lambda_l^t\})||^2 + \frac{\eta^2 L}{2} (\frac{nr + n - 2}{d(n+2)} + 1)\mathbb{E}[||\nabla_{\mathbf{x}} L_p(\mathbf{x}^t, \mathbf{y}^t, \{\lambda_l^t\}; \mathcal{B})||^2].
\end{aligned}
\tag{30}
$$

It follows that

$$
\begin{aligned}
&\mathbb{E}[L_p(\mathbf{x}^{t+1}, \mathbf{y}^t, \{\lambda_l^t\})] - L_p(\mathbf{x}^t, \mathbf{y}^t, \{\lambda_l^t\}) \\
&\leq -\eta ||\nabla_{\mathbf{x}} L_p(\mathbf{x}^t, \mathbf{y}^t, \{\lambda_l^t\})||^2 + \frac{\eta^2 L \gamma}{2} \mathbb{E}[||\nabla_{\mathbf{x}} L_p(\mathbf{x}^t, \mathbf{y}^t, \{\lambda_l^t\}; \mathcal{B})||^2],
\end{aligned}
\tag{31}
$$

where $\gamma = \Theta(r/d) > 1$.

Similar to Eq.(31), according to the descent lemma for stochastic gradient descent (Malladi et al., 2023), we can obtain that

$$
\mathbb{E}[L_p(\mathbf{x}^{t+1}, \mathbf{y}^{t+1}, \{\lambda_l^t\})] - L_p(\mathbf{x}^{t+1}, \mathbf{y}^t, \{\lambda_l^t\})
$$

$$
\leq -\eta||\nabla_{\mathbf{y}} L_p(\mathbf{x}^{t+1}, \mathbf{y}^t, \{\lambda_l^t\})||^2 + \frac{\eta^2 L}{2}\mathbb{E}[||\nabla_{\mathbf{y}} L_p(\mathbf{x}^{t+1}, \mathbf{y}^t, \{\lambda_l^t\}; \mathcal{B})||^2] \tag{32}
$$

$$
\leq -\eta||\nabla_{\mathbf{y}} L_p(\mathbf{x}^{t+1}, \mathbf{y}^t, \{\lambda_l^t\})||^2 + \frac{\eta^2 L\gamma}{2}\mathbb{E}[||\nabla_{\mathbf{y}} L_p(\mathbf{x}^{t+1}, \mathbf{y}^t, \{\lambda_l^t\}; \mathcal{B})||^2],
$$

and

$$
\mathbb{E}[L_p(\mathbf{x}^{t+1}, \mathbf{y}^{t+1}, \{\lambda_1^{t+1}, \lambda_2^t, \cdots, \lambda_{l-1}^t, \lambda_l^t\})] - L_p(\mathbf{x}^{t+1}, \mathbf{y}^{t+1}, \{\lambda_1^t, \lambda_2^t, \cdots, \lambda_{l-1}^t, \lambda_l^t\})
$$

$$
\leq -\eta||\nabla_{\lambda_1} L_p(\mathbf{x}^{t+1}, \mathbf{y}^{t+1}, \{\lambda_1^t, \lambda_2^t, \cdots, \lambda_{l-1}^t, \lambda_l^t\})||^2
$$

$$
+\frac{\eta^2 L\gamma}{2}\mathbb{E}[||\nabla_{\lambda_1} L_p(\mathbf{x}^{t+1}, \mathbf{y}^{t+1}, \{\lambda_1^t, \lambda_2^t, \cdots, \lambda_{l-1}^t, \lambda_l^t\}; \mathcal{B})||^2], \tag{33}
$$

$$
\mathbb{E}[L_p(\mathbf{x}^{t+1}, \mathbf{y}^{t+1}, \{\lambda_1^{t+1}, \lambda_2^{t+1}, \cdots, \lambda_{l-1}^t, \lambda_l^t\})] - L_p(\mathbf{x}^{t+1}, \mathbf{y}^{t+1}, \{\lambda_1^{t+1}, \lambda_2^t, \cdots, \lambda_{l-1}^t, \lambda_l^t\})
$$

$$
\leq -\eta||\nabla_{\lambda_2} L_p(\mathbf{x}^{t+1}, \mathbf{y}^{t+1}, \{\lambda_1^{t+1}, \lambda_2^t, \cdots, \lambda_{l-1}^t, \lambda_l^t\})||^2
$$

$$
+\frac{\eta^2 L\gamma}{2}\mathbb{E}[||\nabla_{\lambda_2} L_p(\mathbf{x}^{t+1}, \mathbf{y}^{t+1}, \{\lambda_1^{t+1}, \lambda_2^t, \cdots, \lambda_{l-1}^t, \lambda_l^t\}; \mathcal{B})||^2], \tag{34}
$$

$\cdots\cdots$

$$
\mathbb{E}[L_p(\mathbf{x}^{t+1}, \mathbf{y}^{t+1}, \{\lambda_1^{t+1}, \lambda_2^{t+1}, \cdots, \lambda_{l-1}^{t+1}, \lambda_l^t\})] - L_p(\mathbf{x}^{t+1}, \mathbf{y}^{t+1}, \{\lambda_1^{t+1}, \lambda_2^{t+1}, \cdots, \lambda_{l-1}^t, \lambda_l^t\})
$$

$$
\leq -\eta||\nabla_{\lambda_{l-1}} L_p(\mathbf{x}^{t+1}, \mathbf{y}^{t+1}, \{\lambda_1^{t+1}, \lambda_2^{t+1}, \cdots, \lambda_{l-1}^t, \lambda_l^t\})||^2
$$

$$
+\frac{\eta^2 L\gamma}{2}\mathbb{E}[||\nabla_{\lambda_{l-1}} L_p(\mathbf{x}^{t+1}, \mathbf{y}^{t+1}, \{\lambda_1^{t+1}, \lambda_2^{t+1}, \cdots, \lambda_{l-1}^t, \lambda_l^t\}; \mathcal{B})||^2], \tag{35}
$$

$$
\mathbb{E}[L_p(\mathbf{x}^{t+1}, \mathbf{y}^{t+1}, \{\lambda_1^{t+1}, \lambda_2^{t+1}, \cdots, \lambda_{l-1}^{t+1}, \lambda_l^{t+1}\})] - L_p(\mathbf{x}^{t+1}, \mathbf{y}^{t+1}, \{\lambda_1^{t+1}, \lambda_2^{t+1}, \cdots, \lambda_{l-1}^{t+1}, \lambda_l^t\})
$$

$$
\leq -\eta||\nabla_{\lambda_l} L_p(\mathbf{x}^{t+1}, \mathbf{y}^{t+1}, \{\lambda_1^{t+1}, \lambda_2^{t+1}, \cdots, \lambda_{l-1}^{t+1}, \lambda_l^t\})||^2
$$

$$
+\frac{\eta^2 L\gamma}{2}\mathbb{E}[||\nabla_{\lambda_l} L_p(\mathbf{x}^{t+1}, \mathbf{y}^{t+1}, \{\lambda_1^{t+1}, \lambda_2^{t+1}, \cdots, \lambda_{l-1}^{t+1}, \lambda_l^t\}; \mathcal{B})||^2]. \tag{36}
$$

For $\mathbf{x}$ variable, according to Eq.(31), denote the step size of the stochastic gradient descent version of our algorithm as $\eta'$, and set $\eta = \frac{\eta'}{\gamma}$, it follows that

$$
\mathbb{E}[L_p(\mathbf{x}^{t+1}, \mathbf{y}^t, \{\lambda_l^t\})] - L_p(\mathbf{x}^t, \mathbf{y}^t, \{\lambda_l^t\})
$$

$$
\leq \frac{1}{\gamma}[-\eta'||\nabla_{\mathbf{x}} L_p(\mathbf{x}^t, \mathbf{y}^t, \{\lambda_l^t\})||^2 + \frac{\eta'^2 L}{2}\mathbb{E}[||\nabla_{\mathbf{x}} L_p(\mathbf{x}^t, \mathbf{y}^t, \{\lambda_l^t\}; \mathcal{B})||^2]]. \tag{37}
$$

Then, set $\eta' \leq \frac{1}{L}$ to have

$$
\mathbb{E}[L_p(\mathbf{x}^{t+1}, \mathbf{y}^t, \{\lambda_l^t\})] - L_p(\mathbf{x}^t, \mathbf{y}^t, \{\lambda_l^t\})
$$

$$
\leq \frac{1}{\gamma}[-\frac{\eta'}{2}||\nabla_{\mathbf{x}} L_p(\mathbf{x}^t, \mathbf{y}^t, \{\lambda_l^t\})||^2 + \frac{\eta'^2 L}{2B}\text{tr}(\Sigma_{\mathbf{x}}(\mathbf{x}^t, \mathbf{y}^t, \{\lambda_l^t\}))]. \tag{38}
$$

For any $w$ in $(\mathbf{x}, \mathbf{y}, \{\lambda_l\})$, following (Malladi et al., 2023), we assume that there exist $\alpha$ such that $\text{tr}(\Sigma_{\boldsymbol{w}}(\mathbf{x}, \mathbf{y}, \{\lambda_l\})) \leq \alpha(L_p(\mathbf{x}, \mathbf{y}, \{\lambda_l\}) - L_p^*)$. Then we have

$$
\mathbb{E}[L_p(\mathbf{x}^{t+1}, \mathbf{y}^t, \{\lambda_l^t\})] - L_p(\mathbf{x}^t, \mathbf{y}^t, \{\lambda_l^t\})
$$

$$
\leq \frac{1}{\gamma}(-\eta'\mu + \frac{\eta'^2 L\alpha}{2B})(\mathbb{E}[L_p(\mathbf{x}^t, \mathbf{y}^t, \{\lambda_l^t\})] - L_p^*) \tag{39}
$$

$$
\Rightarrow \mathbb{E}[L_p(\mathbf{x}^{t+1}, \mathbf{y}^t, \{\lambda_l^t\})] - L_p^* \leq (1 - \frac{1}{\gamma}(\eta'\mu - \frac{\eta'^2 L\alpha}{2B}))(\mathbb{E}[L_p(\mathbf{x}^t, \mathbf{y}^t, \{\lambda_l^t\})] - L_p^*).
$$

Set $\eta' = \min\{\frac{1}{L}, \frac{\mu B}{L\alpha}\}$ to have

$$\mathbb{E}[L_p(\mathbf{x}^{t+1}, \mathbf{y}^t, \{\lambda_l^t\})] - L_p^* \leq \rho(\mathbb{E}[L_p(\mathbf{x}^t, \mathbf{y}^t, \{\lambda_l^t\})] - L_p^*), \tag{40}$$

where $\rho = (1 - \frac{1}{\gamma}(\min\{\frac{\mu}{2L}, \frac{\mu^2 B}{2L\alpha}\}))$.

Similar to $\mathbf{x}$ variable, by analyzing $\mathbf{y}$ and $\{\lambda_l\}$ variables in the same way as Eq.(37), Eq.(38), Eq.(39), and Eq.(40), we can obtain that

$$\mathbb{E}[L_p(\mathbf{x}^{t+1}, \mathbf{y}^{t+1}, \{\lambda_l^t\})] - L_p^* \leq \rho(\mathbb{E}[L_p(\mathbf{x}^{t+1}, \mathbf{y}^t, \{\lambda_l^t\})] - L_p^*),$$

$$\mathbb{E}[L_p(\mathbf{x}^{t+1}, \mathbf{y}^{t+1}, \{\lambda_1^{t+1}, \lambda_2^t, \cdots, \lambda_{l-1}^t, \lambda_l^t\})] - L_p^* \leq \rho(\mathbb{E}[L_p(\mathbf{x}^{t+1}, \mathbf{y}^{t+1}, \{\lambda_1^t, \lambda_2^t, \cdots, \lambda_{l-1}^t, \lambda_l^t\})] - L_p^*),$$

$$\mathbb{E}[L_p(\mathbf{x}^{t+1}, \mathbf{y}^{t+1}, \{\lambda_1^{t+1}, \lambda_2^{t+1}, \cdots, \lambda_{l-1}^t, \lambda_l^t\})] - L_p^* \leq \rho(\mathbb{E}[L_p(\mathbf{x}^{t+1}, \mathbf{y}^{t+1}, \{\lambda_1^{t+1}, \lambda_2^t, \cdots, \lambda_{l-1}^t, \lambda_l^t\})] - L_p^*),$$

$$\cdots\cdots$$

$$\mathbb{E}[L_p(\mathbf{x}^{t+1}, \mathbf{y}^{t+1}, \{\lambda_1^{t+1}, \lambda_2^{t+1}, \cdots, \lambda_{l-1}^{t+1}, \lambda_l^t\})] - L_p^* \leq \rho(\mathbb{E}[L_p(\mathbf{x}^{t+1}, \mathbf{y}^{t+1}, \{\lambda_1^{t+1}, \lambda_2^{t+1}, \cdots, \lambda_{l-1}^t, \lambda_l^t\})] - L_p^*),$$

$$\mathbb{E}[L_p(\mathbf{x}^{t+1}, \mathbf{y}^{t+1}, \{\lambda_1^{t+1}, \lambda_2^{t+1}, \cdots, \lambda_{l-1}^{t+1}, \lambda_l^{t+1}\})] - L_p^* \leq \rho(\mathbb{E}[L_p(\mathbf{x}^{t+1}, \mathbf{y}^{t+1}, \{\lambda_1^{t+1}, \lambda_2^{t+1}, \cdots, \lambda_{l-1}^{t+1}, \lambda_l^t\})] - L_p^*). \tag{41}$$

Combining Eq.(40) and Eq.(41), in the $t+1$ iteration we have

$$\mathbb{E}[L_p(\mathbf{x}^{t+1}, \mathbf{y}^{t+1}, \{\lambda_l^{t+1}\})] - L_p^* \leq \rho^{p+2}(\mathbb{E}[L_p(\mathbf{x}^t, \mathbf{y}^t, \{\lambda_l^t\})] - L_p^*). \tag{42}$$

Denoting $\rho^{p+2}$ as $\rho'$ and according to Eq.(42), we can obtain that

$$\mathbb{E}[L_p(\mathbf{x}^t, \mathbf{y}^t, \{\lambda_l^t\})] - L_p^* \leq \rho'^t(\mathbb{E}[L_p(\mathbf{x}^0, \mathbf{y}^0, \{\lambda_l^0\})] - L_p^*). \tag{43}$$

W can therefor obtain a solution with $\mathbb{E}[L_p(\mathbf{x}^t, \mathbf{y}^t, \{\lambda_l^t\})] - L_p^* \leq \epsilon$ after

$$t = \frac{\gamma}{p+2} \max(\frac{2L}{\mu}, \frac{2L\alpha}{\mu^2 B}) \log(\frac{L_p(\mathbf{x}^0, \mathbf{y}^0, \{\lambda_l^0\}) - L_p^*}{\epsilon})$$

$$= \mathcal{O}((\frac{r}{d} + 1)(\frac{1}{p})(\frac{L}{\mu} + \frac{L\alpha}{\mu^2 B}) \log \frac{L_p(\mathbf{x}^0, \mathbf{y}^0, \{\lambda_l^0\}) - L_p^*}{\epsilon}). \tag{44}$$

$\square$

## A.2    DETAILED EXPERIMENTAL SETTINGS

### A.2.1    NATURAL LANGUAGE UNDERSTANDING TASKS

**Datasets.** For the text classification task, we use the following datasets: 1) SST-2 (The Stanford Sentiment Treebank) (Socher et al., 2013) is used to predict the sentiment of a given sentence in the movie reviews domain. 2) MRPC (The Microsoft Research Paraphrase Corpus) (Dolan & Brockett, 2005) contains pairs of SENTENCE with manual annotations indicating whether the SENTENCE in each pair are semantically equivalent. 3) Tweets_Hate_speech_detection (Lhoest et al., 2021) aims to detect hate speech in tweets. We will abbreviate this dataset as "Tweets_Hate". 4) Wiki_Toxic dataset comprises comments gathered from Wikipedia forums, categorized into two groups: toxic and non-toxic. 5) FELM (Factuality Evaluation of large Language Models) (Chen et al., 2023b) aims to check whether the answer is correct for a question. 6) BoolQ (Wang et al., 2019)is a question answering dataset for yes/no questions. 7) WiC (Wang et al., 2019) is a dataset for word sense disambiguation. Note that the Tweets_Hate and Wiki_Toxic datasets may contain potentially harmful text.

For the multiple choice task, we use COPA (Wang et al., 2019) and SWAG (Zellers et al., 2018) dataset. COPA (The Choice Of Plausible Alternatives) is designed to evaluate open-domain commonsense causal reasoning questions. SWAG (Situations With Adversarial Generations) is a large scale dataset for natural language inference and commonsense reasoning. Finally, for the single-turn question answering task, SQuAD (Rajpurkar et al., 2016) and DROP (Dua et al., 2019) are used. SQuAD (Stanford Question Answering Dataset) (Rajpurkar et al., 2016) is a reading comprehension dataset with questions based on Wikipedia articles. DROP (Discrete Reasoning Over Paragraphs) is a comprehension benchmark requiring discrete reasoning over paragraphs.

Table 7: Prompt Templates for The Cloud-hosted LLM on NLU Tasks.

| Dataset | Templates for the cloud-hosted LLM |
|---|---|
| SST-2 | How is the sentiment of sentence: [OPTIMIZED_INFO]? First respond ONLY with "Great" or "Terrible", then give some explanation. |
| MRPC | Whether [SENTENCE1] and [SENTENCE2] in the pair are semantically equivalent? Note: [OPTIMIZED_INFO]. First respond ONLY with "Yes" or "No", then give some explanation. |
| Tweets_Hate | Whether [OPTIMIZED_INFO] has a racist or sexist sentiment associated with it? First respond ONLY with "Yes" or "No", then give some explanation. |
| Wiki_Toxic | Whether the comment gathered from Wikipedia forums [OPTIMIZED_INFO] is toxic. First respond ONLY with "Yes" or "No", then give some explanation. |
| FELM | For [QUESTION], whether [ANSWER] is a correct answer? Note: [OPTIMIZED_INFO]. First respond ONLY with "Yes" or "No", then give some explanation. |
| BoolQ | Please answer the [QUESTION] based on the [PASSAGE]. Note: [OPTIMIZED_INFO]. First respond ONLY with "Yes" or "No", then give some explanation. |
| WiC | Determine whether the intended sense of the [TEXT] is the same in [SENTENCE1] and [SENTENCE2]. Note: [OPTIMIZED_INFO]. First respond ONLY with "Yes" or "No", then give some explanation. |
| COPA | Choose one from the following two SENTENCE and deduce which sentence is the [QUESTION] of [PREMISE]. Option one: [SENTENCE1]; Option two: [SENTENCE2]. Note: [OPTIMIZED_INFO]. First respond ONLY with "One" or "Two", then give some explanation. |
| SWAG | Choose one from the following four SENTENCE to deduce which sentence might be the end of [SENTENCE0]. Option one: [SENTENCE1]; Option two:[SENTENCE2]; Option three:[SENTENCE3]; Option four: [SENTENCE4]. Note: [OPTIMIZED_INFO]. First respond ONLY with "One" or "Two", "Three", or "Four", then give some explanation. |
| SQuAD/DROP | Please answer the QUESTION and give some explanation. Context Info: [OPTIMIZED_INFO]. Your response should follow the following format: "Answer: ...; Explanation: ...". |

**Prompt Templates.** The prompt templates for the cloud-hosted LLM are summarized in Table 7, where "OPTIMIZED_INFO" denotes the prompts optimized by the edge agent.

**Baselines.** We compare the proposed framework, sandwiched tuning, with the following baselines. 1) Manual Prompt uses the manual designed prompt templates similar to Table 7, but without the OPTIMIZED_INFO. 2) Zero-shot CoT (Kojima et al., 2022) adds a hint, "Let's think step-by-step", on the basis of manual prompt. 3) Random In-Context Learning (ICL) provides a few randomly selected example inputs and their corresponding outputs to guide the model in understanding the context and the type of response. 4) OPRO (Yang et al., 2023) uses an LLM to generate and evaluate new solutions based on the prompt step-by-step.

**Implementation details.** For the edge agent, we employ the low-rank adaptation (LoRA) method for the parameter-efficient fine-tuning of the edge LLM while performing a full-parameter fine-tuning of the adapter model. We use AdamW as the optimizer and set $\eta = 0.0001$. For each dataset,

we use 500 training samples and 50 testing samples. We repeat the experiment on each dataset 5 times and record the average performance.

### A.2.2 MULTI-TURNS DIALOGUE GENERATION

**Datasets.** For datasets, we utilize six customer support datasets, each derived from Twitter interactions, including Hulu_Support, Sainsburys, Comcastcares, Sprintcare, UPSHelp and XboxSupport. Each dataset contains multi-turn dialogues where customers reach out to companies with issues or questions, and support agents respond with resolutions or further queries. These datasets provide a comprehensive view of typical customer support scenarios, covering a range of industries such as entertainment, retail, telecommunications, logistics, and gaming. This variety allows for an in-depth analysis of conversational patterns and the effectiveness of support responses across different sectors.

**Baselines.** For baselines, we compare 2 different strategies for selecting in-context examples:

- Random: Randomly selects dialogue samples without specific optimization.
- ICL: Retrieves 5 dialogues and randomly selects 2 from them for generation.

**Metrics.** For evaluation metrics, we use the "Win Rate" metric, as described by Dubois et al. (2024). The "Win Rate" metric measures how often a dialogue generation method outperforms another in producing higher-quality conversations. In the evaluation process, qwen-max compares two generated dialogues and determines which one is closer to the ground truth. Essentially, it reflects the percentage of times one method's output is judged to be superior to another's in terms of dialogue quality. In our experiments, we use qwen-max's output without any context samples as the competitor.

### A.2.3 TOOL USE TASKS

**Datesets**. For datasets, We use a publicly available mathematical word problem dataset for the mathematical reasoning task(Zhao et al., 2020). Additionally, we created three specialized datasets for floating-point arithmetic, floating-point comparison, and character counting, as shown in Table 8.

To assess the model's performance in floating-point calculation scenarios, we developed the "Float-Arithmetic" dataset, which features real-world problems such as shopping, weighing, and financial calculations. This dataset consists of 500 entries generated by GPT-4o, which were manually verified for accuracy and further calibrated using ChatGLM4 to ensure the reliability of results.

For the floating point comparison task, we built a dataset named Float-Comparison , addressing discrepancies observed in LLM's calculations compared to calculator ground truths. Using LLM (qwen-max), we generated a set of comparison questions based on these results. In the Character Counting task, we created the "Character-Counting" dataset, where the goal is to count occurrences of a specific character in a string, using LLM-generated templates.

**Prompt Templates.** The prompt templates for the cloud-hosted LLM are summarized in Table 10, where "OPTIMIZED_INFO" denotes the prompts optimized by the edge agent. The origin INFO is shown in Table 9.

**Implementation Details.** For the edge agent, we utilize the low-rank adaptation (LoRA) method to perform parameter-efficient fine-tuning on the edge LLM, while applying full-parameter fine-tuning on the adapter model. We use AdamW as the optimizer with a learning rate of $\eta = 0.0001$. For APE-210K dataset, we random select 1000 training samples and 200 testing samples.For another three datasets, each contains 400 training samples and 100 testing samples. We conduct the experiments five times for each dataset and report the average performance results.

**Experiment on tradeoffs among cloud-edge load distribution, inference latency, and inference accuracy.** To demonstrate the system's flexibility in balancing real-time performance and accuracy, we include an additional experiment that dynamically distributes loads between the cloud and

Table 8: Dataset Examples.

| Dataset | Question Example | Answer Example |
|---|---|---|
| Float-Arithmetic | A car rental company charges a daily fee of 45.50 and an additional charge of 0.25 per mile driven. If a customer rents a car for 3 days and drives 150 miles, how much will the total cost be? | 174.0 |
| Float-Arithmetic | A car rental company charges a base fee of 35 per day, with an additional cost of 0.15 per mile driven. If a customer rents a car for 3 days and drives it for 120 miles, how much does the total cost for the rental come to? | 123.0 |
| Float-Arithmetic | You are planning a road trip across three states, and you need to calculate the total cost of fuel. You know the following information: - Your car's average fuel efficiency is 25.7 miles per gallon. - The total distance of the trip is 1,345.6 miles. - Fuel prices vary by state: $3.89 per gallon in the first state for 400 miles, $4.15 per gallon in the second state for 600 miles, and $3.95 per gallon in the third state for the remaining distance. What is the total cost of fuel for your trip? | 210.55 |
| Float-Comparison | Does 58.4 or 58.10 have the upper hand in value? | 58.4 |
| Float-Comparison | Between 49.7 and 49.30, which value is greater? | 49.7 |
| Character-Counting | how many 'i' in word 'kiwifruit'? | 3 |

Table 9: Prompt Templates for the Edge Agent on Tool Use.

| Dataset | Templates for the Edge Agent |
|---|---|
| APE-210K | Note: In a conversational context, when calculations are required, express the entire calculation using a single formula: 'Calculate(expression)'. For example, for 9.10 * 2.5 + 1.23 - 9.8, output: 'Calculate(9.10 * 2.5 + 1.23 - 9.8)'. The 'Calculate(expression)' should encompass the entire calculation process. |
| Float-Arithmetic | Note: In a conversational context, when calculations are required, express the entire calculation using a single formula: 'Calculate(expression)'. For example, for 9.10 * 2.5 + 1.23 - 9.8, output: 'Calculate(9.10 * 2.5 + 1.23 - 9.8)'. The 'Calculate(expression)' should encompass the entire calculation process. |
| Float-Comparison | Rephrase the task as a direct comparison.For example,convert into a sentence like "You need to compare A and B", where A and B are the two numbers to be compared. |
| Character-Counting | Let us think step by step. |

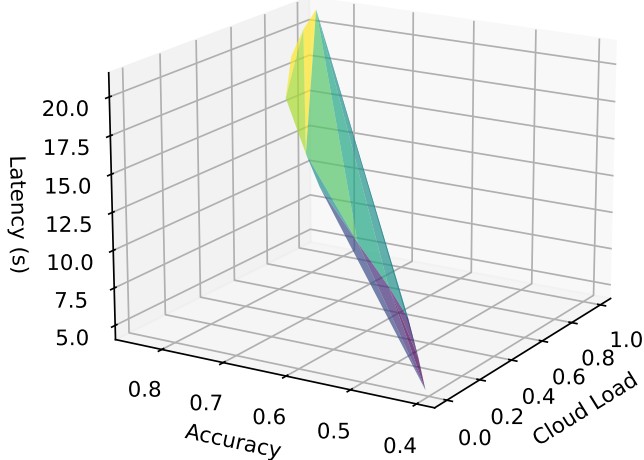

Figure 2: Impact of Cloud-Edge Load Distribution on System Latency and Accuracy.

edge based on query complexity. Specifically, we simulate the cloud-hosted LLM's load using a dataset comprising both complex and simple queries. The complex queries are routed to the cloud-hosted LLM, while the simpler ones are handled by the edge LLM agent. The cloud load in this experiment refers to the proportion of queries assigned to the cloud-hosted LLM.

Table 10: Prompt Templates for the Cloud-hosted LLM on Tool Use.

| Dataset | Templates for the Cloud-Hosted LLM |
|---|---|
| APE-210K | Given the math problem:[QUESTION], Note:[OPTIMIZED INFO]. For all other content, respond normally. |
| Float-Arithmetic | Given the math problem:[QUESTION], Note:[OPTIMIZED INFO]. For all other content, respond normally. |
| Float-Comparison | Given the question:[QUESTION]. Info: [OPTIMIZED INFO]. Response should follow the format: "Answer[sentence]". |
| Character-Counting | Given the question:[QUESTION]. Info: [OPTIMIZED INFO]. Response should follow the format: "Answer[sentence]". |

We analyze the system's overall latency and accuracy under different load distributions. As shown in Figure 2, there is a positive correlation between the load assigned to the cloud-hosted LLM and both latency and accuracy. Notably, reducing the cloud-side load significantly decreases the latency, while the accuracy remains relatively unaffected. This suggests that the edge LLM agent can effectively handle less complex queries, allowing for efficient load balancing between the cloud and edge components.

### A.2.4 LLM TASK DECOMPOSITION

**Datasets.** For the LLM task decomposition task, we use the following datasets: 1) Orca-Math 200K contains approximately 200K grade school math word problems.(Mitra et al., 2024) 2) TaskLAMA (Task Language Model Analysis) is used for testing various task decomposition and measuring the performance (Yuan et al., 2024).

**Prompt Templates.** The prompt templates for the cloud-hosted LLM and edge agents are summarized in Table 12, 13 where "OPTIMIZED_INFO" denotes the prompts optimized by the edge agent.

**Baselines.** We used three large language models with different parameter sizes, namely GPT2, qwen2-7B, and llama3-8B, and compared the task performance before and after optimization.

**Implementation details.** For evaluation metrics, we use F1 score and cosine similarity, as described by (Yuan et al., 2024). F1 score reflects the likelihood that the model can correctly perform task decomposition and cosine similarity assesses the similarity between the task decomposition results of the edge agent and those of the cloud-hosted LLM. In the evaluation process, for the same complex task, both the cloud-hosted LLM and the edge agent perform a task decomposition. The performance of the edge agent's task decomposition is then evaluated based on the results from the cloud-hosted LLM, generating result of cosine similarity and F1 score. Based on these evaluation results, the edge agent is optimized.

Table 11: Prompt Templates for Edge Agent on Tool Use.

| Dataset | Templates for Edge Agent |
|---|---|
| APE-210K and Float-Arithmetic | Your task is to extract a text call to the calculator API, with the output format being 'Calculate(expression)', where "expression" is used for expressions involving +, -, *, and / operators. Only return calls for the specified methods Here are some examples of API calls: Input: To find the area of the tabletop with a cutout, subtract the cutout's area (length x width) from the full tabletop area (length x width). Calculate(2.75 * 1.5 - 0.5 * 0.3) Output: Calculate(2.75 * 1.5 - 0.5 * 0.3) Input: To determine the total cost of the rental, we need to calculate the cost of the miles driven and add it to the base fee. The formula for the total cost is: Total Cost=120+(0.25×150.5), Now, let's express this calculation using the requested format: Calculate(120 + (0.25 * 150.5)) Output: Calculate(120 + (0.25 * 150.5)) Input: [QUESTION] Output:. |
| Float-Comparison | Your task is to add calls to a API named "Compare" to a piece of text. The calls should help you compare two numbers to determine which one is larger. You can call the API by writing "[Compare(number1,number2)]" where number1 and number2 are two numbers needed to be compared.
Examples: - Input: Which is larger, 56.1 or 56.13? Output: Answer:[Compare(56.1, 56.13)]
- Input: Between 993.32 and 993.9, which has the numerical advantage?. Output: Answer:[Compare(999.32, 993.9)]
- Input: Determine the larger number between 78.9 and 78.91. Output: Answer:[Compare(78.9, 78.91)]
- Input: You need to compare 88.11 and 88.3 to determine which one is larger. Output: Answer:[Compare(88.11, 88.3)]
Task: Given the following question, add the 'Compare' calling text and format the output as specified like Answer:[Compare(A,B)].
Input: question Output: |
| Character-Counting | Your task is to add calls to a API named "Count" to a piece of text. The calls should help you count how many chars in a word. You can call the API by writing "[Count(word,char)]",.
Examples: - Input: how many 'r' in word 'kiwifruit'? Output: Answer:[Count(kiwifruit,r)]
- Input: how many 'a' in word 'apricot'? Output: Answer:[Count(apricot,a)]
- Input: how many 'b' in word 'broccoli'? Output: Answer:[Count(broccoli,b)]
Task: Given the following question, add the 'Count' calling text and format the output as specified like Answer:[Count(A,B)].
Input: question Output: |

Table 12: Prompt Templates for the Cloud-hosted LLM on LLM Task Decomposition.

| Dataset | Templates for the cloud-hosted LLM |
| --- | --- |
| Orca-Math 200K | given the math problem:[QUESTION]. Decompose the problem. |
| TaskLAMA | given the problem:[QUESTION]. Decompose the problem. |

Table 13: Prompt Templates for Edge Agent on LLM Task Decomposition.

| Dataset | Templates for edge agent |
| --- | --- |
| Orca-Math 200K | given the math problem:[QUESTION], Note:[OPTIMIZED INFO]. Decompose the problem. |
| TaskLAMA | given the problem:[QUESTION], Note:[OPTIMIZED INFO]. Decompose the problem. |