# OpenReview forum: "Grey-box Prompt Optimization and Fine-Tuning for Cloud-Edge LLM Agents"
_ICLR.cc/2025/Conference — ICLR 2025 Conference Withdrawn Submission_

### Official Review · Reviewer_Xa8r · 2024-10-27

**Soundness:** 3
**Presentation:** 2
**Contribution:** 3
**Rating:** 5
**Confidence:** 2

**Summary:**

This paper introduces a grey-box joint prompt optimization and fine-tuning framework for large-scale LLMs, incorporating the advantages of both cloud-side LLMs and edge-side LLMs. The necessity of considering privacy on the edge-LLM side and the performance/scalability of server-side LLMs is a timely and important topic. The paper’s design is mathematically solid and clear.

**Strengths:**

- Privacy on the edge-LLM side and the performance/scalability of server-side LLMs are timely and important topics
- The paper’s design is mathematically clear

**Weaknesses:**

- The evaluation uses one A100 and two RTX 4090 GPUs. Which serves as the cloud and which as the edge? Also, is it practical for an edge device to be equipped with such GPUs?
- As the authors describe the technique as a “framework,” I suggest they include a clear explanation of a systematic deployment scenario with a figure. Although the algorithm is explained clearly, presenting the end-to-end workflow with framework/system architecture would help readers better understand the high-level concept.
- Using an adapter for server-device collaborative training is a good idea. In this context, there are related studies in split-federated learning for LLMs (e.g., R-SFLLM (arXiv 2024), SplitLora (arXiv 2024), not listed all). These studies share a common goal with this work. Stating the differences between them will improve the research scope

**Questions:**

I have the following questions, including the ones in weakness

- For the cloud side, only one model was used in the experiment. Is there a specific reason, or is there any confidence that different models would not yield different results?
- Can I check the scalability of the proposed ZoCP algorithm about model scale or token length?
- Although previous collaborative LLM research on edge-cloud setups focused on inference, as the authors mentioned, can’t they be partially compared to this technique in the experiments?
- In the design, can you clarify the considerations regarding privacy?

---

### Official Review · Reviewer_N8LG · 2024-11-01

**Soundness:** 2
**Presentation:** 1
**Contribution:** 2
**Rating:** 3
**Confidence:** 5

**Summary:**

The paper proposes a framework for grey-box prompt optimization and fine-tuning aimed at optimizing LLM prompt generation and performance tuning in cloud-side hybrid architectures.

**Strengths:**

Experimental results show that this approach has some performance gains over the traditional approach in several tasks.

**Weaknesses:**

The main contributions of the paper appear to lack significant innovation compared to existing work. The proposed hybrid cloud-edge LLM framework, while emphasizing privacy and performance through edge deployment, largely reiterates concepts already established in the field. The cloud-edge framework is not novel in itself, and the introduction of grey-box optimization only presents minor advancements over existing black-box and white-box optimization techniques, rather than a ground-breaking new methodology. And the evaluation primarily focuses on prompt optimization and fine-tuning without a comprehensive assessment of other important aspects of cloud-edge LLM systems, such as communication overhead, data synchronization issues, and latency impacts, which are critical factors in edge-cloud architectures. And also lacks some figures to enhance the presentation.

**Questions:**

The proposed Sandwiched Tuning and Zeroth-Order Cutting Plane algorithm may introduce significant computational complexity, particularly for large-scale deployments. The paper does not provide a detailed analysis of the scalability of these methods when implemented on limited-resource edge devices.

---

### Official Review · Reviewer_kmPc · 2024-11-04

**Soundness:** 2
**Presentation:** 2
**Contribution:** 2
**Rating:** 5
**Confidence:** 3

**Summary:**

The paper proposes a framework for integrating cloud and edge LLMs to address challenges related to privacy, scalability, and computational efficiency. The authors introduce a "Sandwiched Tuning" approach, which utilizes grey-box optimization to enable joint prompt optimization and fine-tuning. It relies on  a memory-efficient Zeroth-Order Cutting Plane (ZoCP) algorithm that guarantees non-asymptotic convergence, making it suitable for edge deployments. The authors claim performance improvements (up to 47.9%) over traditional methods across a range of tasks, including task decomposition, tool use, multi-turn dialogue generation, and natural language understanding.

**Strengths:**

- The authors is tackling an improtant problem in collaboration of cloud and edge models.
- The paper is generally well-structured, with claiming its contributions, problem formulation, and experimental results.
- An extensive tasks and datasets are employed for evaluation.

**Weaknesses:**

- The paper can be significantly improved for clarity if including more details in methodolody and implementation of experiments. Several key details to evaluate the contribution of the paper seem to be missing. E.g. What is the functionality of the adaptor model and how does it "enhance generation" of the edge model? What is the specific process for fine-tuning the edge and adapter models? How does prompt optimization work in conjunction with the fine-tuning process?

 - While the paper includes a general comparison to cloud-edge LLM collaboration frameworks does not adequately compare the proposed method to other methods that also fine-tune the edge model. And thus, it is difficult for the audiance to determine the novelty and new contribution of the paper.

**Questions:**

Please also refer to my questions in "weakness" section.

- In section 4.1, the adapter is not mentioned. Is it not used for this experiment?

- I am not sure if I understand where the latency improvement come for Sandwiched Tuning compared to cloud only. It seems that the cloud model need to be called anyways.

- How does this approach compared to other fine-tuned only approach for edge models?

---

### Official Review · Reviewer_1Y5y · 2024-11-04

**Soundness:** 2
**Presentation:** 3
**Contribution:** 2
**Rating:** 5
**Confidence:** 3

**Summary:**

Based on the lens of grey-box optimization, this paper introduces an LLM agent framework, which is called Sandwiched Tuning, for fine-tuning large language models (LLMs) in hybrid cloud-edge environments, balancing performance, data privacy, and resource efficiency. The core of this framework is a memory-efficient Zeroth-Order Cutting Plane (ZoCP) algorithm for edge deployment, enabling secure, personalized fine-tuning directly on edge devices. The non-asymptotic convergence rate of the ZoCP algorithm has been rigorously derived. Extensive experiments show great performance gains over traditional methods.

**Strengths:**

● The paper is clearly structured.
● Rigorous proof for the non-asymptotic convergence of the proposed algorithm has been provided.
● Comprehensive experimental data for validating the effectiveness of the proposed framework has been given, including comparison between Sandwiched Tuning and baselines in four different tasks, and the ablation study on different components.
● The proposed framework demonstrates significant performance improvement over the baselines on some tasks.

**Weaknesses:**

● The first contribution of this paper is the introduction of the Cloud-Edge LLM agent framework. The related work section mentions that there has been a lot of research focusing on a collaborative framework consisting of cloud-based and edge LLMs. However, this paper does not explicitly explain how the framework differs from previous works. In the methodology section, the authors directly discuss the ZoCP algorithm without explaining the exact process of Sandwiched Tuning or providing related diagrams that can help readers understand the differences. Therefore, the novelty of the framework remains doubtful. In addition, the authors exaggerate the advantages of the framework in introduction section but do not explain these advantages in detail and poor evidence are provided to justify these credits. For example, the summarized contribution in introduction mentions that the approach optimises performance, maintains privacy and improves resource management, but only the result of optimising performance is reflected throughout the paper. The problem setup introduced in the methodology session also does not cover factors such as privacy, resource cost, and communication bandwidth.
● The paper highlights that the proposed framework poses a huge improvement in performance over the baselines. However, Section 4.2 shows that the improvement actually comes from the comparison with cloud LLM frameworks and edge LLM frameworks, not with the same type of frameworks. The specific settings of the experiments mentioned in the main body and appendix do not reflect whether this comparison is fair. If the proposed framework that uses both cloud and edge LLMs uses more computing resources, then outperforming the baselines that only fine-tune the cloud-based or edge LLMs does not justify that the proposed method is superior.
● Similarly, although Table 5 shows that the proposed method outperforms the baseline method on the NLU task, it is still unclear whether the selected baselines use the same LLMs and whether they train with both cloud and edge LLMs as the proposed method. The provided implementation details in main body and appendix does not mention this. Based on the scaling laws of LLMs, if the baseline method uses LLMs with smaller model sizes, then the performance degradation is actually caused by the model aspect rather than the algorithm aspect.
● Furthermore, Table 4 compares the sandwiched tuning with the Random and ICL baselines. However, both baselines are based on random selection in fact and do not correspond to the performance of the latest research work. The comparison with the two overly naive baseline yields a flawed performance advantage.

**Questions:**

Regarding the problem and algorithm formulation in Section 3, there are some parts requiring further clarification:
1. In equation 1, since the outer problem is to optimize the model parameters x of the edge LLM, should it be argmin_x F(x, y) rather than min_x F(x,y)?
2. The zero-order optimization in Eqs.4 and 12 introduces an hyperparameter \epsilon. How does the value of this parameter affect the training?
3. Generally, the first-order Taylor approximation cannot reveal the complete nature of the objective function. Why can it be justified to be applied in this case? The expansion at higher order also holds significant uses for finding the optimal solution.
4. For the cutting plane updating mechanism, combining Eq.14 and Eq.15 seems not guarantee that h(x,y) <= \epsilon, which is the required constraint in Eq.5. If this is correct, then why is the found closed-form solution of the new cutting plane able to help formulating the feasible set?

---

> ### Comment · Reviewer_1Y5y · 2024-11-21
> **I maintain my rating**
>
> Thank you for your detailed response. Based on the replies, I decided to maintain my score for this paper, as I think the paper would require too much additional content to address the issues, and many of my concerns still have not been addressed by the replies. Below are my further comments:
>
> 1. (W1) For the weakness 1, my main concern is the unclear contributions, as there is no direct explanation on how this hybrid framework differs from the existing works and no direct evidence justifying the other privacy and resource management merits. Authors' replies show that they are willing to add a new figure to show their framework and additional descriptions to highlight the advantages. However, I think that these changes that should be added would drastically alter the structure of the original submission. For example, having additional sections to explain why this framework is innovative and how these advantages are verified. This is a difficult task requiring more consideration, so the solutions provided by authors at this stage do not make me satisfied.
> 2. (W2-W3) My other main concern is whether the comparison experiments are performed in a fair setup. Since the innovation of this framework is the collaboration between cloud and edge LLMs, then it is highly important that there is a fair comparison to frameworks using single LLM. If the proposed framework does use two times the resources over the baselines due to utilizing multiple LLMs, than the performance improvement does not hold sufficient novelty. Unfortunately, the reply from the authors still not address this concern. Although the authors seem to emphasize their contributions and the models used by baselines, I do not get the specific point of whether this is a fair comparison from the perspectives of computation and resources.
> 3. (W4) In addition to the above two points, I am also concerned about the selection of the baselines. The paper considers significant performance improvement as one of the key contributions, so it is reasonable to compare the performance of the proposed method to some state-of-the-art baselines in Table 4. If the results of the two current baselines are the only ones that can be provided at this stage, I do not agree that it is enough to well justify that the proposed method holds a significant performance advantage over the previous methods on this task.
> 4. (Questions) Thanks for the clarification to my questions. I have been more clear about the paper, but there are still some problems.
>     - (Q2) According to the reply, it looks like the value of $\epsilon$ is really important for the proposed method. Then why there is no experiments to test its impact on the performance? It is necessary to provide this missing result.
>     - (Q3) These additional justifications about the use of the first-order Taylor expression should also be provided in the original manuscript. Additionally, it is still not clear why the higher-order terms could be ignored here. If the main reason is that the first-order term provides better efficiency, then there should also be some theoretical or empirical evidence to validate this point.

---

### Note · Authors · 2024-11-29

I have read and agree with the venue's withdrawal policy on behalf of myself and my co-authors.